# COARSE-TO-FINE CONCEPT DISCOVERY MODELS: A CONCEPT PYRAMID SCHEME

## ABSTRACT

Deep Learning algorithms have recently gained significant attention due to their impressive performance. However, their high complexity and un-interpretable mode of operation hinders their confident deployment in real-world safety-critical tasks. This work targets *ante hoc* interpretability, and specifically Concept Bottleneck Models (CBMs). Our goal is to design a framework that admits a highly interpretable decision making process with respect to human understandable concepts, on *two levels of granularity*. To this end, we propose a novel hierarchical concept discovery formulation leveraging: (i) recent advances in image-text models, and (ii) an innovative formulation for *coarse-to-fine concept selection* via data-driven and sparsity inducing Bayesian arguments. Within this framework, concept information does not solely rely on the similarity between the *whole* image and general unstructured concepts; instead, we introduce the notion of *concept hierarchy* to uncover and exploit more granular concept information residing in patch-specific regions of the image scene. As we experimentally show, the proposed construction not only outperforms recent CBM approaches, but also yields a principled framework towards interpetability.

## 1 INTRODUCTION

The recent advent of multimodal models has greatly popularized the deployment of Deep Learning approaches to a variety of tasks and applications. However, in most cases, deep architectures are treated in an alarming *black-box* manner: given an input, they produce a particular prediction, with their mode of operation and complexity preventing any potential investigation of their decision-making process. This property not only raises serious questions concerning their deployment in safety-critical applications, but at the same time it could actively preclude their adoption in settings that could otherwise benefit societal advances, e.g., medical applications.

This conspicuous *shortcoming* of modern architectures has fortunately gained a lot of attention from the research community in recent years, expediting the design of novel frameworks towards Deep Neural Network (DNN) interpretability. Within this frame of reference, there exist two core approaches: *ante-* and *post-* hoc. The latter aims to provide *explanations* to conventional pretrained models, e.g., Network Dissection (Bau et al., 2017), while the former aims to devise *inherently* interpretable models. In this context, Concept Bottleneck Models (CBMs) constitute one of the best-known approaches; these comprise: (i) an intermediate Concept Bottleneck Layer (CBL), a layer whose neurons are tied to human understandable *concepts*, e.g., textual descriptions, followed by (ii) a linear decision layer. Thus, the final decision constitutes a linear combination of the CBL's concepts, leading to a more interpretable decision mechanism. However, typical CBM approaches are accompanied by four significant drawbacks: (i) they commonly require hand-annotated concepts, (ii) they usually exhibit lower performance compared to their non-interpretable counterparts, (iii) their interpretability is substantially impaired due to the sheer amount of concepts that need to be analysed during inference, and (iv) they are not suited for tasks that require greater granularity.

The first drawback has been recently addressed by incorporating image-text models in the CBM pipeline; instead of relying on a fixed concept set, any text can be projected in the image-text embedding space and compared with the image. At the same time, mechanisms to restore performance have also been proposed, e.g., residual fitting (Yuksekgonul et al., 2022). The remaining two limitations however, still pose a significant research challenge.

Indeed, CBMs usually rely on a large amount of concepts, usually proportional to the number of classes for the given task; with more complex datasets, thousands of concepts may be considered. Evidently, this renders the investigation of the decision making tasks an *arduous* and *unintuitive* process. In this context, some works aim to reduce the amount of considered concepts by imposing sparsity constraints upon concept activation. Commonly, post-hoc class-wise sparsity methods are considered (Wong et al., 2021; Oikarinen et al., 2023); however, these tend to restrict the number of concepts on a *per-class* basis, enforcing *ad hoc* application-specific sparsity/performance thresholds, greatly limiting the flexibility of concept activation for each example. Recently, a data-driven per-example discovery mechanism has been proposed in Panousis et al. (2023); this leverages binary indicators founded upon Variational Bayesian arguments and explicitly denote the relevance of each concept on a per-example basis. This allows for a greater flexibility, since each example can activate a number of concepts that have been deemed essential to achieve the downstream task.

Even though these approaches aim address the problem of concept over-abundance, they do not consider ways to emphasize finer concept information that may present in a given image; they still exclusively target similarity between concepts and the *whole image*. In this setting, localized, low-level concepts (e.g. object shape or texture), are predicted from a representation of the whole image, potentially leading to the undesirable use of top-down relations. For instance, the model detects some high-level concept (e.g., elephant), resulting in associated lower-level concept activations (e.g., tusks, wrinkled skin) that may not even be actually be visible. This can further lead to significant concept omission, i.e., information potentially crucial for tasks that require greater granularity, e.g., fine-grained part discovery, or even cases where the input is susceptible to multiple interpretations.

Drawing inspiration from this inadequacy of CBM formulations, we introduce a novel coarse-to-fine paradigm that allows for discovering and capturing both *high* and *low* level concept information. We achieve this objective by: (i) leveraging recent CBM advances, namely Concept Discovery Models (CDMs), (ii) devising an end-to-end trainable hierarchical construction; in this setting, we exploit both the whole image, as well as information residing in individual isolated regions of the image, i.e., specific patches, to achieve the downstream task. These levels of hierarchy are linked together by intuitive and principled arguments, allowing for information and context sharing between them, paving the way towards more interpretable models. We dub our approach *Concept Pyramid Models* (CPMs); in principle, our framework allows for arbitrarily deep hierarchies using different representations, e.g., super-pixels. Here, we focus on the two-level setting, as a proof of concept for the potency of the proposed framework. Our contributions can be summarized as follows:

- We introduce a novel interpretable hierarchical model that allows for coarse-to-fine concept discovery, exploiting finer details residing in patch-specific regions of an image.
- We propose a novel way of assessing the interpretation capacity of our model based on the Jaccard index between ground truth concepts and learned data-driven binary indicators.
- We perform a thorough quantitative and qualitative analysis. We experimentally show that CPMs outperform other SOTA approaches classification-wise, while substantially improving interpretation capacity.

## 2 RELATED WORK

CBMs decompose the final task of prediction into multiple concept detection tasks, allowing for a richer evaluation of the model's reasoning. Early works on concept-based models (Mahajan et al., 2011), were severely limited by requiring an extensive hand-annotated dataset comprising all the used concepts. In this context, and to enhance the reliability of predictions of diverse visual contexts, probabilistic approaches, such as ProbCBM(Kim et al., 2023), build upon conventional CBMs, introducing the concept of *ambiguity*, allowing for capturing the uncertainty both in concept and class prediction. The appearance of image-text models, chiefly CLIP (Radford et al., 2021), has mitigated the need for hand-annotated data, allowing to easily make use of thousands of concepts, followed by a linear operator on the concept presence probabilities to solve the downstream task (Oikarinen et al., 2023; Yang et al., 2023b). However, this generally means that all concepts may simultaneously contribute to a given prediction, rendering the analysis of concept contribution an arduous and unintuitive task, severely undermining the sought-after interpetability. This has led to methods that seek also a sparse concept representation, either by design (Marcos et al., 2020) or data-driven (Panousis et al., 2023), which is the approach we follow in this work.

## 3 CONCEPT PYRAMID MODELS

Let us denote by $\mathcal{D} = \{\boldsymbol{X}_n, \hat{\boldsymbol{y}}_n\}_{n=1}^N$, a dataset comprising $N$ images, where each image $\boldsymbol{X}_n \in \mathbb{R}^{I_H \times I_W \times c}$ comprises $c$ channels, and $\hat{\boldsymbol{y}}_n \in \{0, 1\}^C$ its class label. Within the context of CBMs, a *concept set* $\mathbb{A} = \{a_1, \dots, a_H\}$, comprising $H$ concepts, e.g., textual descriptions, is also considered; the main objective is to re-formulate the prediction process, constructing a *bottleneck* that relies upon the considered concepts, in an attempt to design inherently interpretable models. In this work, we deviate from the classical definition of CBMs and consider the setting of *coarse-to-fine* concept-based classification based on similarities between images and concepts.

**Concept-based Classification.** To discover the relations between images and attributes, image-language models, and specifically CLIP (Radford et al., 2021), are typically considered. These comprise an image and a text encoder, denoted by $E_I(\cdot)$ and $E_T(\cdot)$ respectively, trained in a contrastive manner (Sohn, 2016; Chen et al., 2020) to learn a common embedding space. After training, we can then project any image and text in this common space and compute the similarity between their ($\ell_2$-normalized) embeddings. Thus, assuming a concept set $\mathbb{A}$, with $|\mathbb{A}| = H$, the most commonly considered similarity measure $\boldsymbol{S}$ is the cosine similarity:

$$\boldsymbol{S} \propto E_I(\boldsymbol{X}) E_T(\mathbb{A})^T \in \mathbb{R}^{N \times H} \tag{1}$$

This *similarity-based representation* has recently been exploited to design models with interpretable decision processes such as CBM-variants (Yuksekgonul et al., 2022; Oikarinen et al., 2023) and Network Dissection approaches (Oikarinen & Weng, 2023). Evidently, the similarity $\boldsymbol{S}$ yields a unique representation for each image and can directly be used towards downstream tasks.

Let us consider a $C$-class classification setting; by introducing a linear layer $\boldsymbol{W}_c \in \mathbb{R}^{H \times C}$, we can perform classification via the similarity representation $\boldsymbol{S}$. The output of such a network yields:

$$\boldsymbol{Y} = \boldsymbol{S} \boldsymbol{W}_c^T \in \mathbb{R}^{N \times C} \tag{2}$$

In this setting, the image and text encoders are usually kept frozen, and training only pertains to the weight matrix $\boldsymbol{W}_c$. This approach has been shown to yield impressive results despite the simplicity of the approach and even on low-resolution datasets such as CIFAR-10 (Panousis et al., 2023).

However, this simple formulation comes with a key deficit: it is by-design limited to the granularity of the concepts that it can potentially discover in any particular image. Indeed, for any given image, image-text models are commonly trained to match *high-level concepts* present therein; this leads to a *loss of granularity*, that is, important details in the image are either omitted or considered irrelevant. Yet, in complex tasks such as fine-grained classification or in cases where the decision is ambiguous, this can potentially hinder both the downstream task, but also interpretability. In these settings, it is likely that any low-level information present is not captured, obstructing any potential low-level investigation on how the network reasoned on the high-level concept. Moreover, this approach considers the *entire concept set* to describe an input; this not only greatly limits the flexibility of the considered framework, but also renders the interpretation analyses questionable due to the sheer amount of concepts that need to be analysed during inference (Ramaswamy et al., 2023).

In this work, we consider a novel hierarchical concept discovery formulation, introducing the notion of *hierarchy* of concepts, represented by two distinct yet dependent modeling *levels*: *High (H)* and *Low (L)*. To this end, we introduce: (i) the high level concepts $\mathbb{A}_H$; each concept therein is characterized by a number of attributes, thus forming the (ii) low-level pool of concepts (attributes) $\mathbb{A}_L$. The former are used to discover an image's concept representation in the context of the *whole* image, while the latter are used to uncover finer information residing in patch-specific regions. Each considered level aims to achieve the given downstream task, while information sharing takes place between them as we describe in the following.

### 3.1 HIGH LEVEL CONCEPT DISCOVERY

For the high-level, we consider: (i) the whole image, and (ii) the set of $H$ concepts $\mathbb{A}_H$. Using the definitions of concept-based classification, i.e. Eqs.(1), (2), we can perform classification using a single linear layer with weights $\boldsymbol{W}_{Hc} \in \mathbb{R}^{H \times C}$:

$$\boldsymbol{S}_H \propto E_I(\boldsymbol{X}) E_T(\mathbb{A}_H)^T \in \mathbb{R}^{N \times H} \tag{3}$$

$$\boldsymbol{Y}_H = \boldsymbol{S}_H \boldsymbol{W}_{Hc}^T \in \mathbb{R}^{N \times C} \tag{4}$$

In this formulation however, all the considered concepts are potentially contributing to the final decision, not taking into account the relevance of each concept towards the downstream task or any information redundancy; simultaneously, the interpretation capacity is also limited due to the large amount of concepts that need to be analysed during inference. To bypass this drawback, we consider a novel, data-driven mechanism for concept discovery based on auxiliary *binary* latent variables.

**Concept Discovery.** To discover the *essential subset* of high-level concepts to represent each example, we introduce appropriate auxiliary binary latent variables $\mathbf{Z}_H \in \{0,1\}^{N \times H}$; these operate in an "on"-"off" fashion, indicating, for each example, if a given concept needs to be considered to achieve the downstream task, i.e., $[\mathbf{Z}_H]_{n,h} = 1$ if concept $h$ is *active* for example $n$, and $0$ otherwise. The output of the network is now given by the inner product between the classification matrix $\mathbf{W}_{Hc}$ and the *effective concepts* as dictated by the binary indicators $\mathbf{Z}_H$:

$$\mathbf{Y}_H = (\mathbf{Z}_H \cdot \mathbf{S}_H)\mathbf{W}_{Hc}{}^T \in \mathbb{R}^{N \times C} \qquad (5)$$

A naive definition of these indicators would require computing and storing one indicator per example. To avoid the computational complexity and generalization limitations of such a formulation, we consider an *amortized* approach similar to (Panousis et al., 2023). To this end, we introduce a data-driven random sampling procedure for $\mathbf{Z}_H$, and postulate that the latent variables are drawn from appropriate Bernoulli distributions; specifically, their probabilities are proportional to a separate linear computation between the *embedding of the image* and an *auxiliary linear layer* with weights $\mathbf{W}_{Hs} \in \mathbb{R}^{K \times M}$, where $K$ is the dimensionality of the embedding, yielding:

$$q([\mathbf{Z}_H]_n) = \text{Bernoulli}\left([\mathbf{Z}_H]_n \Big| \text{sigmoid}\left(E_I(\mathbf{X}_n)\mathbf{W}_{Hs}{}^T\right)\right) \in \{0,1\}^H, \quad \forall n \qquad (6)$$

where $[\cdot]_n$ denotes the $n$-th row of the matrix, i.e., the indicators for the $n$-th image. This formulation exploits an *additional source of information* emerging solely from the image embedding; this allows for an *explicit* mechanism for inferring concept relevance in the context of the considered task, instead of exclusively relying on the *implicit* CLIP similarity measure. However, considering only the high-level concept information can be insufficient, since it potentially ignores the effect of any fine-grained details present in an image. To this end, we introduce a novel low-level concept discovery mechanism that is then directly tied to the described high-level formulation.

## 3.2 LOW LEVEL CONCEPT DISCOVERY

For formulating a finer concept discovery mechanism, we introduce the notion of *concept hierarchy*. Specifically, we assume that each of the $H$ high-level concepts is characterized by a number of low-level attributes; these are pooled together to form the set of $L$ low-level concepts $\mathbb{A}_L$. In general, high-level concepts may or may not share any low-level attributes. Within this framework, reusing the whole image may hinder concept discovery since fine-grained details may be ignored in the context of the whole image. Moreover, prominent objects may dominate the discovery task, especially in complex scenes, while other significant attributes present in different regions of the image can be completely be ignored.

Thus, to facilitate the discovery of low-level information, avoiding conflicting information in the context of whole image, we split each image $n$ into a set of $P$ *non-overlapping* patches: $\mathbf{P}_n = \{\mathbf{P}_n^1, \mathbf{P}_n^2, \ldots, \mathbf{P}_n^P\}$, where $\mathbf{P}_n^p \in \mathbb{R}^{P_H \times P_W \times c}$ and $P_H, P_W$ denote the height and width of each patch respectively, and $c$ is the number of channels. In this context, each patch is now treated as a standalone image. To this end, we first compute the similarities with respect to the pool of low-level concepts. For each image $n$ split into $P$ patches, the patches-concepts similarity computation reads:

$$[\mathbf{S}_L]_n \propto E_I(\mathbf{P}_n)E_T(\mathbb{A}_L)^T \in \mathbb{R}^{P \times L}, \quad \forall n \qquad (7)$$

We define a single classification layer with weights $\mathbf{W}_{Lc} \in \mathbb{R}^{L \times C}$, while for obtaining a single representation vector for each image, we introduce an *aggregation* operation to combine the information from all the patches. This can be performed before or after the linear layer. Here, we consider the latter, using a maximum rationale. Thus, for each image $n$, the output $[\mathbf{Y}_L]_n \in \mathbb{R}^C$, reads:

$$[\mathbf{Y}_L]_n = \max_p \left[[\mathbf{S}_L]_n \mathbf{W}_{Lc}^T\right]_p \in \mathbb{R}^C, \quad \forall n \qquad (8)$$

where $[\cdot]_p$ denotes the $p$-th row of the matrix. This formulation still exhibits the same issue as the simple concept-based approach: all low-level concepts are potentially considered, hindering the

interpretation process. To this end, we define the corresponding concept discovery mechanism for the low level to address information redundancy and then introduce an information linkage between the different levels.

**Concept Discovery.** For each patch $p$ of image $n$, we consider latent variables $[\boldsymbol{Z}_L]_{n,p} \in \{0,1\}^L$, operating in an "on"-"off" fashion as before. Specifically, we introduce an amortization matrix $W_{Ls} \in \mathbb{R}^{K \times L}$, $K$ being the dimensionality of the embeddings. In this setting, $[\boldsymbol{Z}_L]_{n,p}$ are drawn from Bernoulli distributions driven from the patch embeddings, s.t.:

$$q([\boldsymbol{Z}_L]_{n,p}) = \text{Bernoulli}\left([\boldsymbol{Z}_L]_{n,p}\big|\text{sigmoid}\left(E_I([\boldsymbol{P}]_{n,p})\boldsymbol{W}_{Ls}^T\right)\right) \in \{0,1\}^L, \quad \forall n, p \quad (9)$$

The output is now given by the inner product between the *effective low level concepts* as dictated by $\boldsymbol{Z}_L$ and the weight matrix $\boldsymbol{W}_{Lc}$, yielding:

$$[\boldsymbol{Y}_L]_n = \max_p \left[\left([\boldsymbol{Z}_L]_n \cdot [\boldsymbol{S}_L]_n\right)\boldsymbol{W}_{Lc}^T\right]_p \in \mathbb{R}^C, \forall n \quad (10)$$

The formulation of the low-level, patch-focused variant is now concluded. This can be used as a standalone network to uncover information residing in patch-specific regions of an image and investigate the network's decision making process. However, we can further augment this functionality by linking the two described levels, allowing the flow of information between them.

### 3.3 LINKING THE TWO LEVELS

For tying the two different levels together, we exploit: (i) the latent variables $\boldsymbol{Z}_H, \boldsymbol{Z}_L$, and (ii) the relationship between the high and low level concepts. Since for each high-level concept we have access to which concepts from the low-level pool of attributes characterizes it, we can use this information for context exchange between the two levels.

Specifically, for each high-level concept $h$, we consider a *fixed* $L$-sized binary vector $\boldsymbol{b}_h \in \{0,1\}^L$ that encodes its relationship with the attributes; these are concatenated to form the matrix $\boldsymbol{B} \in \{0,1\}^{L \times H}$. Each entry $l, h$ therein, denotes if the low-level attribute $l$ characterizes the high-level concept $h$; if so, $[\boldsymbol{B}]_{l,h} = 1$, otherwise $[\boldsymbol{B}]_{l,h} = 0$. It is important to highlight that we do not require any ground truth information for constructing $\boldsymbol{B}$; its construction is solely based on the concept sets. However, if ground-truth indicators denoting the relation between high and low level concepts is available, we can easily exploit it as prior information.

Constructing $\boldsymbol{B}$ is a very intuitive process. For example consider the high-level concept *cat* and a pool of attributes [*fur, paws, bricks, eggs, tail*]. In this setting, $\boldsymbol{b}_{\text{cat}} = [1, 1, 0, 0, 1]$, since we expect a *cat* to be characterized by *fur, paws* and *tail*, and not by *bricks* and *eggs*. Hence, we can *mask* the low-level concepts, and zero-out the ones that are irrelevant, following a top-down rationale. During training, we learn which high-level concepts are active, and subsequently discover the relevance of low-level attributes, while the probabilistic nature of our construction allows for the consideration of different configurations of high and low level concepts. This leads to a rich information exchange between the high and the low levels of the network towards achieving the downstream task. A discussion of the top-down and bottom-up rationale of concept hierarchy is provided in the Appendix.

To formalize this linkage, we first consider which high-level concepts are active via $\boldsymbol{Z}_H$ and $\boldsymbol{B}$ to uncover which low-level attributes should be considered in the final decision; this is computed via a mean operation, averaging over the high-level dimension $H$. Then, we use the indicators $\boldsymbol{Z}_L$ to further mask the remaining low-level attributes. This yields:

$$\boldsymbol{Z} \propto \left(\boldsymbol{Z}_H \boldsymbol{B}^T\right) \cdot \boldsymbol{Z}_L \quad (11)$$

Thus, by replacing the indicators $\boldsymbol{Z}_L$ in Eq.10 with $\boldsymbol{Z}$, the two levels are linked together and can be trained on an end-to-end fashion. A graphical illustration of the proposed Concept Pyramid Models (CPM) is depicted on Fig. 1. The introduced framework can easily accommodate more than two levels of hierarchy, while allowing for the usage of different input representations, e.g., super-pixels.

### 3.4 TRAINING & INFERENCE

**Training.** Considering a dataset $\mathcal{D} = \{(\boldsymbol{X}_n, \hat{\boldsymbol{y}}_n)\}_{n=1}^N$, we employ the standard cross-entropy loss, denoted by $\text{CE}(\hat{\boldsymbol{y}}_n, f(\boldsymbol{X}_n, \boldsymbol{A}))$, where $f(\boldsymbol{X}_n, \boldsymbol{A}) = \text{Softmax}([\boldsymbol{Y}]_n)$ are the class probabilities. For

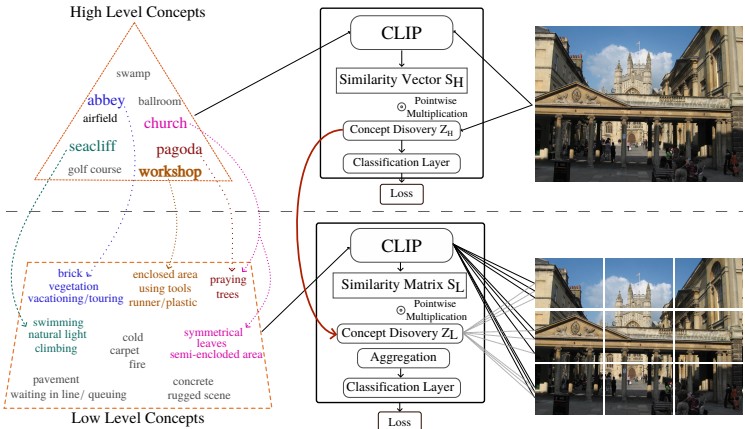

Figure 1: A schematic of the envisioned Concept Pyramid Models. We consider a set of high level concepts, each described by a number of attributes; this forms the *pool* of low-level concepts. Our objective is to discover concepts that describe the whole image, while exploiting information residing in patch-specific regions. To this end, we match low-level concepts to each patch and aggregate the information to obtain a single representation to achieve a downstream task. The levels are tied together via the concept indicators $\boldsymbol{Z}_H, \boldsymbol{Z}_L$ and the relationship between the concepts.

the simple concept-based model, i.e., without any discovery mechanism, the logits $[\boldsymbol{Y}]_n$ correspond to either $[\boldsymbol{Y}_H]_n$ (Eq.(4)), or $[\boldsymbol{Y}_L]_n$ (Eq.(8)), depending on the considered level. In this context, the only trainable parameters are the classification matrices for each level, i.e., $\boldsymbol{W}_{Hc}$ or $\boldsymbol{W}_{Lc}$.

For the full model, the presence of the indicator variables, i.e., $\boldsymbol{Z}_H$ and/or $\boldsymbol{Z}_L$, necessitates a different treatment of the objective. To this end, we turn to the Variational Bayesian (VB) framework, and specifically to Stochastic Gradient Variational Bayes (SGVB) (Kingma & Welling, 2014). We impose appropriate prior distributions on the latent indicators $\boldsymbol{Z}_H$ and $\boldsymbol{Z}_L$, such that:

$$\boldsymbol{Z}_H \sim \text{Bernoulli}(\alpha_H), \qquad \boldsymbol{Z}_L \sim \text{Bernoulli}(\alpha_L) \tag{12}$$

where $\alpha_H$ and $\alpha_L$ are non-negative constants. In the following, we consider the case where the levels are linked together. Obtaining the objective for a single level is trivial; one only needs to remove the other level's terms. Since the network comprises two outputs, the loss function consists of two distinct CE terms: (i) one for the high-level, and (ii) one for the low-level. The final objective function takes the form of an Evidence Lower Bound (ELBO) (Hoffman et al., 2013):

$$
\begin{aligned}
\mathcal{L}_{\text{ELBO}} = \sum_{i=1}^{N} & \varepsilon \text{CE}\big(\hat{\boldsymbol{y}}_n, f(\boldsymbol{X}_n, \boldsymbol{A}_H, [\boldsymbol{Z}_H]_n)\big) + (1-\varepsilon)\text{CE}\big(\hat{\boldsymbol{y}}_n, f(\boldsymbol{X}_n, \boldsymbol{A}_L, [\boldsymbol{Z}]_n)\big) \\
& - \beta\big(\text{D}_{KL}\big(q([\boldsymbol{Z}_H]_n)\big|\big|p([\boldsymbol{Z}_H]_n)\big) + \sum_p \text{D}_{KL}\big(q([\boldsymbol{Z}_L]_{n,p})\big|\big|p([\boldsymbol{Z}_L]_{n,p})\big)\big)
\end{aligned}
\tag{13}
$$

where we augmented the CE notation to reflect the dependence on the binary indicators and $\varepsilon$ is a balancing term. $\beta$ is a scaling factor (Higgins et al., 2017) to avert the KL term from dominating the downstream task. The KL term encourages the posterior to be close to the prior; setting $\alpha_H, \alpha_L$ to a very small value "pushes" the posterior to sparser solutions. Through training, we aim to learn which of these components effectively contribute to the downstream task.

For computing Eq. (13), we turn to Monte Carlo (MC) sampling using a single reparameterized sample for each latent variable. Since, the Bernoulli is not amenable to the reparameterization trick (Kingma & Welling, 2014), we turn to its continuous relaxation using the Gumbel-Softmax trick (Maddison et al., 2017; Jang et al., 2017); we present the exact sampling procedure in the appendix.

**Inference.** After training, we can directly draw samples from the learned posteriors and perform inference. Specifically, let us assume an input image $\boldsymbol{X}$; this is first passed through the high-level discovery mechanism (Eq. (6)), from which we draw samples of the high-level concept indicators $\boldsymbol{Z}_H$ and compute the high-level output based on Eq.(5). We then turn to the low-level: first the image is split into patches. We then draw samples for the patch-specific indicators $\boldsymbol{Z}_L$ according to Eq.(9). We combine the low and the high level information through Eq.(11) and compute the output for the low-level. Finally, apart from assessing the classification capacity, we can investigate the latent indicators on each level to gain insights on the network's decision making process.

## 4    EXPERIMENTAL EVALUATION

**Experimental Setup.**    We consider three different benchmark datasets for evaluating the proposed hierarchical framework, namely, CUB, SUN, and ImageNet-1k. These constitute highly diverse datasets varying in both number of examples and applicability: ImageNet is a 1000-class object recognition benchmark, SUN comprises 717 classes with a limited number of examples for each, while CUB is used for fine-grained bird species identification spanning 200 classes. For the Vision-Language models, we turn to CLIP(Radford et al., 2021) and select a common backbone, i.e., ViT-B/16. To avoid having to calculate the embeddings of both images/patches and text at each iteration, we pre-compute them with the chosen backbone. Then, during training, we directly load them and compute the necessary quantities. For the high level concepts, we consider the class names for each dataset. For the low-level concepts, we consider: (i) for SUN and CUB, the ground-truth attributes comprising 102 and 312 descriptions respectively, and (ii) for ImageNet, we randomly select 20 concepts for each class from the concept set described in Yang et al. (2023a). These distinct sets enables us to assess the efficacy of the proposed framework in highly diverse configurations. For constructing $B$, we consider: (i) for SUN and CUB, a per-class summary stemming from the ground truth relationship between classes and attributes, (ii) for ImageNet, a binary representation of the 20 active entries for each concept. We consider both classification accuracy, as well as the capacity of the proposed framework towards interpretability. For all experiments, we set $\alpha_H, \alpha_L$, and $\beta$ to $1e - 4$ and $\epsilon = 0.5$. Further details can be found in the Appendix.

**Accuracy.**    We begin our experimental analysis by assessing both the classification capacity of the proposed framework, but also its *concept sparsification* ability. To this end, we consider: (i) a baseline non-intepretable backbone, (ii) the recently proposed SOTA Label-Free CBMs (Oikarinen et al., 2023), (iii) classification using only the clip embeddings either of the whole image (CLIP Embeddings[H]) or the image's patches (CLIP Embeddings[L]), (iv) classification based on the similarity between images and the *whole* concept set (CDM[H] ✗discovery), and (v) the approach of Panousis et al. (2023) that considers a data-driven concept discovery mechanism only on the whole image (CDM[H]✓discovery). We also consider the proposed patch-specific variant of CDMs defined in Sec. 3.2, denoted by CDM[L]. The baseline results and the Label-Free CBMs are taken directly from (Oikarinen et al., 2023). We denote our novel hierarchical framework as CPM.

In this setting, models based on the images' patches, i.e. CLIP[L] and CDM[L], are trained with the pool of low-level attributes as concepts. Here, it is worth noting that the CDM[L] setting corresponds to a variant of the full CPM model, where all the high level concepts are active; thus, all attributes are considered in the low-level with no masking involved. However, in this case, since the binary indicators $Z_H$ are not used, there is no information exchange taking place between the two levels; this serves as an ablation setting that allows for assessing the impact of the information linkage.

The obtained comparative results are depicted in Table 1. Therein, we observe that the proposed framework exhibits highly improved performance compared to Label-Free CBMs, while on par or even improved classification performance compared to the concept discovery-based CDMs on the high-level. On the low level, our approach improves performance up to $\approx 20\%$ compared to CDM[L].

At this point, it is important to highlight the effect of the hierarchical construction and the linkage of the levels to the overall behavior of the network. In all the considered settings, we observe: (i) a drastic improvement of the classification accuracy of the low-level module, and (ii) a significant change in the patterns of concept discovery on both levels. We posit that the information exchange that takes place between the levels, conveys a *context* of the relevant attributes that should be considered. This is reflected both to the capacity to improve the low-level classification rate compared to solely using the CLIP embeddings or CDM[L], but also on the drastic change of the concept retention rate of the low level. At the same time, the patch-specific information discovered on the low-level alters the discovery patterns of the high-level, since potentially more concepts should be activated in order to successfully achieve the downstream task. This behavior is extremely highlighted in the ImageNet case: our approach not only exhibits significant gains compared to the alternative concept-based CDM[H] on the high-level, but also the low-level accuracy of our approach *outperforms* it by a large margin. These first investigations hint at the capacity of the proposed framework to exploit patch-specific information for improving on the considered downstream task.

**Attribute Matching.**    Even though classification performance constitutes an important indicator of the overall capacity of a given architecture, it is not an appropriate metric for quantifying its

| Architecture Type | Model | Concepts | Sparsity | Dataset (Accuracy (%) \|\| Sparsity (%)) | | |
|---|---|---|---|---|---|---|
| | | | | CUB | SUN | ImageNet |
| Non-Interpretable | Baseline (Images) | ✗ | ✗ | 76.70 | 42.90 | 76.13 |
| | CLIP Embeddings[H] | ✗ | ✗ | 81.90 | 65.80 | 79.40 |
| | CLIP Embeddings[L] | ✗ | ✗ | 47.80 | 46.00 | 62.85 |
| Concept-Based Whole Image | Label-Free CBMs | ✓ | ✓ | 74.59 | − | 71.98 |
| | CDM[H] | ✓ | ✗ | 80.30 | 66.25 | 75.22 |
| | CDM[H] | ✓ | ✓ | **78.90**\|\|19.00 | **64.55**\|\|13.00 | 76.55\|\|14.00 |
| | CPM[H] (Ours) | ✓ | ✓ | 77.80\|\|42.30 | 64.00\|\|47.58 | **77.40**\|\|27.20 |
| Concept-Based Patches | CDM[L] | ✓ | ✗ | 39.05 | 37.00 | 49.20 |
| | CDM[L] | ✓ | ✓ | 59.62\|\|58.00 | 42.30\|\|67.00 | 58.20\|\|25.60 |
| | CPM[L] (Ours) | ✓ | ✓ | 72.00\|\|24.00 | 57.10\|\|28.33 | 78.45\|\|15.00 |

Table 1: Classification Accuracy and Average Percentage of Activated Concepts (Sparsity). By **bold** black/blue, we denote the best-performing high/low level *sparsity*-inducing concept-based model.

behavior within the context of interpretability. To this end, and contrary to recent approaches that solely rely on classification performance and qualitative analyses, we introduce a metric to measure the effectiveness of a concept-based approach. Thus, we turn to the *Jaccard Similarity* and compute the similarity between the binary indicators $z$ that denote the *discovered* concepts and the binary ground truth indicators that can be found in both CUB and SUN; we denote the latter as $z^{\mathrm{gt}}$.

Let us denote by: (i) $M_{11}$ the number of entries equal to 1 in both binary vectors, (ii) $M_{0,1}$ the number of entries equal to 0 in $z$, but equal to 1 in $z^{\mathrm{gt}}$, and (iii) $M_{1,0}$ the number of entries equal to 1 in $z$, but equal to 0 in $z^{\mathrm{gt}}$; we consider the *asymmetric case*, focusing on the importance of correctly detecting the presence of a concept. Then, we can compute the Jaccard similarity as:

$$\mathrm{Jaccard}(z, z^{\mathrm{gt}}) = M_{1,1}/(M_{1,1} + M_{1,0} + M_{0,1}) \tag{14}$$

The considered metric can be exploited as an objective score for evaluating the quality of the obtained concept-based explanations across multiple frameworks, given they consider the same concept set and the ground truth indicators exist.

For a baseline comparison, we train a CDM with either: (i) the whole image (CDM) or (ii) the image patches (CDM[L]), using the *whole set* of low-level attributes as the concept set for both SUN and CUB. We consider the same set for the low-level of CPMs; due to its hierarchical nature however, CPM can exploit *concept hierarchy* as described in Sec.3.3 to narrow down the concepts considered on the low-level. For both SUN and CUB, we have ground truth attributes on a per-example basis (*example-wise*), but also the present attributes per class (*class-wise*). We assess the matching between these ground-truth indicators and the inferred indicators both in terms of binary accuracy, but also in terms of the considered Jaccard index.

| Model | Attribute Set Train | Atrribute Set Eval | Dataset (Matching Accuracy (%)\|\| Jaccard Index (%)) | |
|---|---|---|---|---|
| | | | SUN | CUB |
| CDM(Panousis et al., 2023) | whole set | class-wise | 51.43\|\|26.00 | 39.00\|\|17.20 |
| CDM(Panousis et al., 2023) | whole set | example-wise | 48.45\|\|15.70 | 36.15\|\|09.50 |
| CDM[L] | whole set | class-wise | 30.95\|\|26.70 | 25.81\|\|19.60 |
| CDM[L] | whole set | example-wise | 20.70\|\|15.00 | 17.65\|\|10.40 |
| CPM (Ours) | hierarchy | class-wise | **53.10**\|\|**28.20** | **79.85**\|\|**27.20** |
| CPM (Ours) | hierarchy | example-wise | 49.92\|\|16.80 | 81.00\|\|16.10 |

Table 2: Attribute matching accuracy. We compare our approach to the recently proposed CDM model trained the considered low-level concept sets. Then, we predict the matching, in terms of Jaccard similarity, between the inferred per-example concept indicators and: (i) the per example and (ii) class-wise ground truth attributes found in both SUN and CUB.

In Table 2, the attribute matching results are depicted. Therein we observe, that our CPMs outperform both CDM and CDM[L] in all the different configurations and in both the considered metrics with up to $10\%$ improvement. These results suggest that by exploiting concept and representation hierarchy, we can uncover low-level information and more relevant concepts. However, it is also important to note how the binary accuracy metric can be quite misleading. Indeed, the ground truth indicators, particularly in CUB, are quite sparse; thus, if a model predicts that most concepts are not relevant, we yield very high binary accuracy. Fortunately though, the proposed metric can successfully address this false sense of confidence as a more appropriate measure for concept matching.

**Qualitative Analysis.** For our qualitative analysis, we focus on the ImageNet-1k validation set; this decision was motivated by the fact that it is the only dataset where attribute matching could not

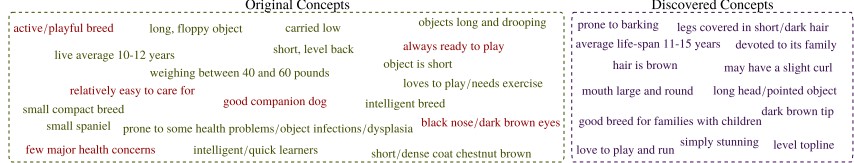

Figure 2: Original and additional discovered concepts for the *Sussex Spaniel* ImageNet class. By green, we denote the concepts retained from the original low-level set pertaining to the class, by maroon, concepts removed via the binary indicators $Z$, and by purple the newly discovered concepts.

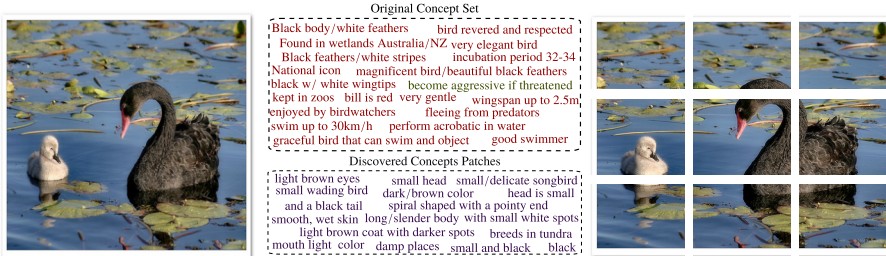

Figure 3: A random example from the *Black Swan* class of ImageNet-1k validation set. On the upper part, the original concept set corresponding to the class is depicted, while on the lower, some of the concepts discovered via our novel patch-specific formulation.

be assessed due to the absence of ground-truth information. Thus, in Fig. 2, we selected a random class (*Sussex Spaniel*) and depict: (i) the 20 originally considered concepts and (ii) the results of the concept discovery. In this setting, we consider a concept to be relevant to the class if it is present in more than $40\%$ of the examples of the class; these concepts are obtained by averaging over the class examples' indicators. We observe that our CPM is able to retain highly relevant concepts from the original set, while discovering equally relevant concepts from other classes such as *australian terrier, soft-coated wheaten terrier* and *collie*.

Finally, in Fig.3, for a random image from the ImageNet-1k validation set, we illustrate: (i) the original set of concepts describing its class (*Black Swan*), and (ii) some of the low-level attributes discovered by our CPM. We observe that the original concept set pertaining to the class cannot adequately represent the considered example. Indeed, most concepts therein would make the intepretation task difficult even for a human annotator. In stark contrast, the proposed framework allows for a more interpretable set of concepts, capturing finer information residing in the patches; this can in turn facilitate a more thorough examination of the network's decision making process.

## 5 LIMITATIONS & CONCLUSIONS

A potential limitation of the proposed framework is the dependence on the pretrained image/text encoders. The final performance and interpretation capacity is tied to the suitability of the backbone with respect to the task at hand. If the embeddings cannot adequately capture the relation (in terms of similarity) between images/patches-concepts, there is currently no mechanism to mitigate this issue. However, if this issue arises, the introduced construction can easily accommodate any suitable modifications by simply altering the embedding networks. Concerning the complexity of the proposed CPM framework, by precomputing all the required embeddings for a considered task, the resulting complexity is orders of magnitude lower than training a conventional backbone.

In this work, we proposed an innovative framework in the context of ante-hoc interpretability based on a novel hierarchical construction. We introduced the notion of *concept hierarchy*, in which, high-level concepts are characterized by a number of lower-level attributes. In this context, we leveraged recent advances in CBMs and Bayesian arguments to construct an end-to-end coarse-to-fine network that can exploit these distinct concept representations, by considering both the whole image, as well as its individual patches; this facilitated the discovery and exploitation of finer information residing in patch-specific regions of the image. We validated our paradigm both in terms of classification performance, while considering a new metric for evaluating the network's capacity towards interpretability. As we experimentally showed, we yielded networks that retain or even improve classification accuracy, while allowing for a more granular investigation of their decision process.

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

# A    APPENDIX

## A.1    THE TOP-DOWN AND BOTTOM-UP VIEW OF CONCEPT DESIGN

As noted in the main text, in this work, we consider both high and low level concepts. Within this frame of reference, one design choice that needed to be addressed is the interpretation of the connection between these two sets. In our view, there are two ways to approach this setting: (i) top-down (which we consider), and (ii) bottom-up. We posit that a combined bottom-up-then-top-down approach is what would most closely follow a human-like behavior when analysing an object. However, it is the second step that is more of a conscious process: we first become aware of the whole object, e.g., a bird or a dog, even if we have subconsciously perceived a lot of low-level cues to reach that conclusion, and then, based on this high-level knowledge, we can draw further conclusions about the nature of the lower-level image characteristics, e.g. interpreting a furry texture as either feathers or fur.

In a purely bottom-up approach, we would first analyse the low-level characteristics, such as shapes and textures, and we would then try to reason about the whole context in order to assign them semantics, e.g. ears, tail, fur. In our opinion, there isn't a single right approach for solving this problem in the context of interpretability. We posit however, that the information exchange that takes places between the high and the low levels via the learning process of the binary indicators does indeed allows for context information sharing between both levels (in the forward pass only from high to low, but also the inverse during training).

One of the motivations of this work was to be able to examine not only the high level concepts but mainly the low level ones. This could potentially allow for drawing conclusions about the high level concept in terms of the uncovered low level attributes. In this context, we can focus on the discovered low-level attributes themselves and reason on the high-level concepts as the reviewer suggests. In our opinion, this is somewhat captured in the proposed framework. Indeed, in the qualitative analyses, we observed that, many times, the discovered low level concepts revealed attributes that are semantically connected to various high level concepts.

**Future work.**    In our setting, we assume that there exists a known hierarchy/relationship between the concepts. However, it may very well be the case that there exists some hidden/latent hierarchy in the ground truth attributes that is not explicitly captured via the construction of the concepts sets. In this case, an interesting extension to our proposed framework would be a compositional bottom-up approach with no a priori known hierarchies. Within this context, we could potentially devise a method that explicitly integrates the aforementioned bottom-up view, aiming to uncover the hidden hierarchies.

## A.2    DISTINCTION BETWEEN THE CPM FRAMEWORK AND VISION TRANSFORMERS

Both our framework and Vision Transformers (ViTs), consider a setting where an image is split into patches, which are then used for learning representations either in the context of conventional non-intepretable settings (ViTs) or via inherently interpretable models (CPMs). However, in ViTs, the patch representations are allows to interact freely with each other along the model, allowing for rich, context-aware representations, but limiting their interpretability, since it becomes impossible to grasp these interactions. At the same time, it is not straightforward to exploit the low-level hidden spatial information in the context of concept-based interpretability in a principled way. In our case, we strongly limit patch interactions by only allowing a single channel of communication that allows the high level concepts to control which low level concepts are detected by providing the appropriate semantic context and aggregate the individual information arising from each patch region.

### A.3 BERNOULLI RELAXATION

To compute the ELBO in Eq.equation 13, we turn to Monte-Carlo sampling, using a single reparameterized sample. Since, the Bernoulli distribution is not amenable to the reparameterization trick (Kingma & Welling, 2014), we turn to its continuous relaxation(Maddison et al., 2017; Jang et al., 2017).

Let us denote by $\tilde{z}_i$, the probabilities of $q(z_i)$, $i = 1, \ldots N$. We can directly draw reparameterized samples $\hat{z}_i \in (0,1)^M$ from the continuous relaxation as:

$$\hat{z}_i = \frac{1}{1 + \exp\left(-(\log \tilde{z}_i + L)/\tau\right)} \tag{15}$$

where $L \in \mathbb{R}$ denotes samples from the Logistic function, such that:

$$L = \log U - \log(1 - U), \quad U \sim \text{Uniform}(0, 1) \tag{16}$$

where $\tau$ is called the *temperature* parameter; this controls the degree of the approximation: the higher the value the more uniform the produced samples and vice versa. We set $\tau$ to $0.1$ in all the experimental evaluations. During inference, we can use the Bernoulli distribution to draw samples and directly compute the binary indicators.

### A.4 EXPERIMENTAL DETAILS

For our experiments, we set $\alpha_H = \alpha_L = \beta = 10^{-4}$; we select the best performing learning rate among $\{10^{-4}, 10^{-3}, 5 \cdot 10^{-3}, 10^{-2}\}$ for the linear classification layer. We set a higher learning rate for $\boldsymbol{W}_{Hs}$ and $\boldsymbol{W}_{Ls}$ ($10\times$) to facilitate learning of the discovery mechanism. We set $\epsilon = 0.5$ to equally balance the contribution of the classification losses. Table 3 provides an ablation study on the effect of this parameter in the final performance. As the value of $\epsilon$ increases, we observe a gradual improvement in the classification performance of both levels. We posit that the classification signal on the high level allows for the discovery of more discriminant features towards classification while restricting the impact of the sparsity inducing mechanism (to achieve the downstream task). Simultaneously, this provides more flexibility to the low level since more high level concepts are considered, leading to both accuracy improvements and a larger number of low level attributes. This leads to a more expressive flow on information between the two levels, exhibiting the final behavior described in the main text.

| $\epsilon$ | 0.0 | 0.1 | 0.2 | 0.5 | 0.7 | 0.8 | 1.0 |
|---|---|---|---|---|---|---|---|
| High Level | 0.430\|\|10.50 | 73.00\|\|23.25 | 76.20\|\|42.50 | 79.20\|\|57.00 | 79.50\|\|58.00 | 79.50\|\|57.00 | 80.50\|\|51.60 |
| Low Level | 62.55\|\|21.10 | 69.50\|\|23.50 | 71.20\|\|30.00 | 72.45\|\|31.00 | 73.00\|\|33.00 | 72.30\|\|31.50 | 0.500\|\|00.00 |

Table 3: Effect of the epsilon parameter on the performance of the two levels. When taking "extreme" values, the respective classification performance of each level collapses since there is no classification signal.

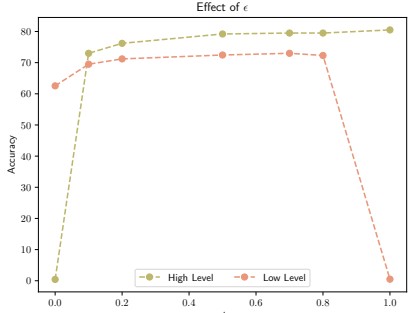
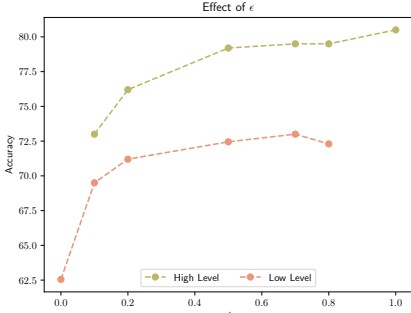

Figure 4: Effect of the $\epsilon$ parameter on the classification performance of the the two levels. **Left**: We observe that when taking "extreme" values, i.e., $\epsilon = 0$ or $\epsilon = 1$ the classification performance of the respective level collapses. **Right**: Visualization without the collapsed accuracies to highlight the effect of $\epsilon$ on the two levels.

For all our experiments, we use the Adam optimizer without any complicated learning rate annealing schemes. We trained our models using a single NVIDIA A5000 GPU with no data parallelization. For all our experiments, we split the images into $3 \times 3$ patches.

For SUN and CUB, we train the model for a maximum of 2000 epochs, while for ImageNet, we only train for 100 epochs.

**Complexity.** For SUN and CUB, training each configuration for 2000 epochs, takes approximately 5 minutes (wall time measurement). On the other hand, for ImageNet, 100 epochs require approximately 4 hours. Computing and saving the embeddings for all datasets requires couple of minutes.

## A.5 FURTHER QUALITATIVE ANALYSES

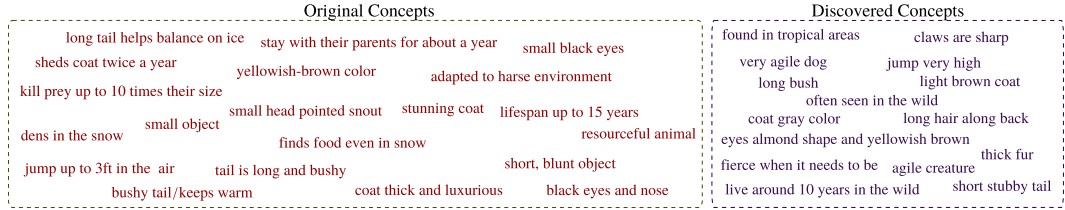

Figure 5: Original and additional discovered concepts for the *Arctic Fox* ImageNet class. By green, we denote the concepts retained from the original low-level set pertaining to the class, by maroon, concepts removed via the binary indicators $Z$, and by purple the newly discovered concepts.

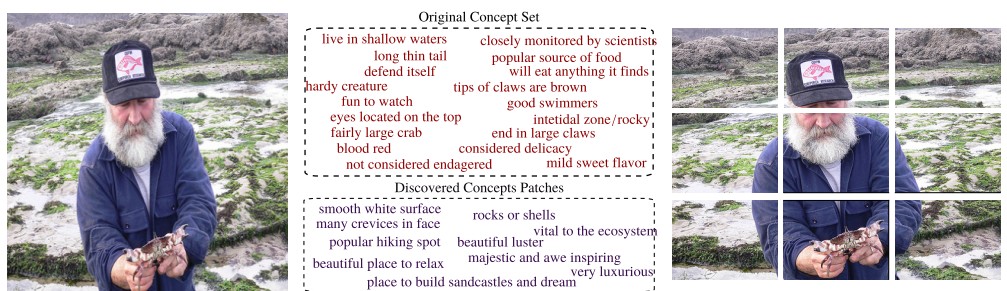

Figure 6: A random example from the *Rock Crab* class of ImageNet-1k validation set. On the upper part, the original concept set corresponding to the class is depicted, while on the lower, some of the concepts discovered via our novel patch-specific formulation.

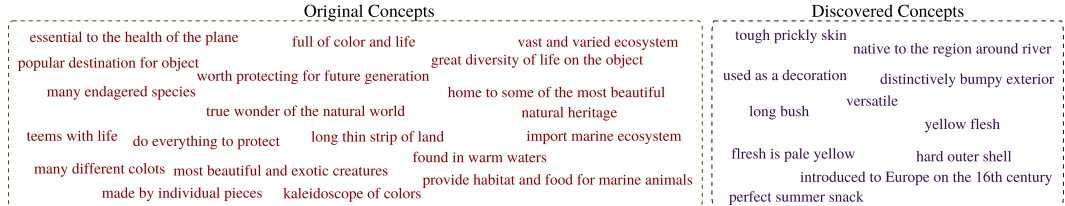

Figure 7: Original and additional discovered concepts for the *Coral Reef* ImageNet class. By green, we denote the concepts retained from the original low-level set pertaining to the class, by maroon, concepts removed via the binary indicators $Z$, and by purple the newly discovered concepts.

