# OpenReview forum: "Hierarchical Concept Discovery Models: A Concept Pyramid Scheme"
_ICLR.cc/2024/Conference — ICLR 2024 Conference Withdrawn Submission_

### Official Review · Reviewer_iVqG · 2023-10-30

**Soundness:** 3 good
**Presentation:** 2 fair
**Contribution:** 2 fair
**Rating:** 3
**Confidence:** 5

**Summary:**

This paper extends the concept bottleneck model (CBM) to develop a hierarchical concept concept discovery formulation. The proposed method leverages recent multimodal models such as CLIP and a multi-level concept selection algorithm. Different from CBMs, the authors claim that the proposed method can discover a hierarchy of concepts.

-----------After Rebuttal--------------

I have read the author response and other reviews. I would like to thank the authors for their response, which is helpful. However, my major concerns remain (e.g., in terms of novelty and the notion of high/low-level concepts). I would therefore keep my ratings unchanged. I would suggest that the authors make a stronger case in terms of where to draw the line between high- and low-level concepts and what their technical contributions are in their revision.

**Strengths:**

The notion of concept hierarchy seems interesting.

The proposed method outperforms an adapted label-free version of CBM.

**Weaknesses:**

The motivation of low-level information does not seem very convincing to me. The authors argue that previous methods such as label-free CBM focus on only image-level concepts and therefore could lose fine-grained details in the image. However, one could also argue that the fine-grained details in the image are already well summarized into the image-level concepts. For example, the CUB dataset does contain a large number of low-level concepts (information) such as color and shape of birds’ parts.

Equation 6 essentially introduces an additional gating mechanism to turn on/off different concepts. The key goal is to introduce sparsity of concept activation. While this make sense to me, why not use a simpler approach, e.g., sending S_H into another linear layer, followed by some L1 regularization? Would that work too?

The proposed method seems to heavily build on Panousis et al., 2023, introducing limited novelty. For example, the whole high-level concept discovery section is nearly identical to Panousis et al., 2023. For the low-level concept discovery, it is actually a simple adaptation, i.e., replacing the whole image in Equation (3-6) with each patch, resulting in Equation (7-9). The only difference is in the aggregation of the patch-level information and the link between both levels.

The idea of using pretrained models like CLIP is not new either, with the baseline label-free CBMs as an example.

The notation is a bit confusing. In Equation (7), does $[P]_n$ refer to the collection of all $P_n$ where $n$ goes from 1 to P? Or does it only refer to the $n$-th patch? Or does it refer to all patches from Image $n$? If it is the former, this is no different from Equation (3). Similar questions arise for Equation (8). What is $c$ in the size of $P_p$? Is it the number of image channels?

It is unclear how to draw the line between high-level concepts and low-level attributes. In Section 3.3 the authors mentioned *cat* as a high-level concept and *eggs* as low-level attributes. Why couldn’t *eggs* be a high-level concept? This also goes back to the motivation of introducing low-level concepts mentioned in the comments above. From Section 4, it seems the so-called *concept* is simply the class label and the low-level concepts are actually the commonly used *concept* in CUB/SUN? This does not make too much sense.

In terms of performance, the proposed CPM underperforms CDM by a considerable margin on both CUB and SUN. Also, it is not fair to claim that label-free CBMs is not capable of concept discovery, since they do automatically discover concepts using CLIP. The only advantage is in terms of sparsity, which can actually be trivially enforced by additional sparsity constraints in label-free CBMs.

**Questions:**

Why not use a simpler approach, e.g., sending S_H into another linear layer, followed by some L1 regularization? Would that work too?

The notation is a bit confusing. In Equation (7), does $[P]_n$ refer to the collection of all $P_n$ where $n$ goes from 1 to P? Or does it only refer to the $n$-th patch? Or does it refer to all patches from Image $n$? If it is the former, this is no different from Equation (3). Similar questions arise for Equation (8). What is $c$ in the size of $P_p$? Is it the number of image channels?

It is unclear how to draw the line between high-level concepts and low-level attributes. In Section 3.3 the authors mentioned *cat* as a high-level concept and *eggs* as low-level attributes. Why couldn’t *eggs* be a high-level concept? This also goes back to the motivation of introducing low-level concepts mentioned in the comments above. From Section 4, it seems the so-called *concept* is simply the class label and the low-level concepts are actually the commonly used *concept* in CUB/SUN?

---

> ### Author Response · Authors · 2023-11-12
>
> We thank the reviewer for his consideration and suggestions/clarification on our work.
>
> > The motivation of low-level information does not seem very convincing to me. The authors argue that previous methods such as label-free CBM focus on only image-level concepts and therefore could lose fine-grained details in the image. However, one could also argue that the fine-grained details in the image are already well summarized into the image-level concepts. For example, the CUB dataset does contain a large number of low-level concepts (information) such as color and shape of birds’ parts.
>
> We thank the reviewer for giving us the opportunity to expand on the benefit of using a separate mechanism to uncover low-level information. One could indeed argue that potentially some low-level information can be captured in the context of the whole image, although this would need to be empirically validated. In this context, let us consider the experimental results depicted in Table 2 and focus on CUB. In this setting, for all methods, we consider the same low-level concepts of CUB (such as the color and shape of bird's parts, as the reviewer suggests). Comparing the performance, in terms of concept detection, of CDM of Panousis et al. (2023), which constitutes a standard CBM variant that uses the whole image, and our CPM method that aims to uncover low-level information, we observe $40$% and $35$% absolute difference when considering the binary matching accuracy and $10$% and $7$% absolute difference in performance using the Jaccard Index. Thus, the experimental results strongly hint at the benefits of the proposed mechanism. We consider the same concept set for all methods, but our method is able to exploit the two levels of information in order to uncover more appropriate low-level information, thus supporting the motivation and premise of the approach.
>
> > Equation 6 essentially introduces an additional gating mechanism to turn on/off different concepts. The key goal is to introduce sparsity of concept activation. While this make sense to me, why not use a simpler approach, e.g., sending S\_H into another linear layer, followed by some L1 regularization? Would that work too?
>
> Indeed, the discovery mechanism is based on an additional gating functionality. $L_1$ regularization alone, without an appropriate mechanism to project the activation values onto a simplex structure, does indeed have sparsity inducing properties and could (in our opinion) produce sparse results. However, the $L_1$ regularization would have to be imposed on the weights of the linear layer; this would turn off concepts completely for all images or on a per-class basis (using more involved methods), requiring at the same time explicit and unintuitive sparsity-accuracy thresholds, severely limiting the flexibility of the approach.
>
> > The proposed method seems to heavily build on Panousis et al., 2023, introducing limited novelty. For example, the whole high-level concept discovery section is nearly identical to Panousis et al., 2023. For the low-level concept discovery, it is actually a simple adaptation, i.e., replacing the whole image in Equation (3-6) with each patch, resulting in Equation (7-9). The only difference is in the aggregation of the patch-level information and the link between both levels.
>
> Indeed, this method builds upon the work of Panousis et al. (2023), which we cite throughout the manuscript. However, as noted by the reviewer, we introduce and assess a variant of the method that not only allows to detect concepts at two levels of granularity, but also to let them interact with each other, as well a novel evaluation in the context of intepretability using the Jaccard index and ground truth information. We honestly believe that we did not overestimate the novelty of our approach since, to the best of our knowledge both the hierarchical construction and the ground-truth evaluation process have not been yet explored in the literature before. At the same time, the experimental results vouch for the efficacy of the proposed method, yielding substantial improvements in attribute matching.
>
> > The idea of using pretrained models like CLIP is not new either, with the baseline label-free CBMs as an example.
>
> While we completely agree with the reviewer that employing a pretrained CLIP model is not new, we would stress again that the main novelty of our work lies in the hierarchical construction leading to a refined concept-based classification and the unique evaluation process.
>
> > The notation is a bit confusing.
>
> We thank the reviewer for raising the issue of the notation. $[P]\_n$ corresponds to the set of patches of the $n$-th image, while $[]_p$ denotes the entry corresponding to the $p$-th patch; $c$ is the number of channels. We will address these issues in the camera-ready version.

---

> ### Author Response · Authors · 2023-11-12
>
> > It is unclear how to draw the line between high-level concepts and low-level attributes. In Section 3.3 the authors mentioned cat as a high-level concept and eggs as low-level attributes. Why couldn’t eggs be a high-level concept? This also goes back to the motivation of introducing low-level concepts mentioned in the comments above. From Section 4, it seems the so-called concept is simply the class label and the low-level concepts are actually the commonly used concept in CUB/SUN? This does not make too much sense.
>
> The construction of the high and low level concepts is related to the task, the dataset or the intention of the implementation.  Eggs could easily be a high level concept, if the application also comprises images of fried eggs; the low-level concepts could be "yolks", "whites", "in a pan". Eggs could also be a high level concept if the low level concepts describe it as "round", "brown or white", "come from chickens", "found in a hen house". The "so-called" concepts can be any textual description that the user intends to use, following a top-down rationale. In all the considered settings, the high-level concepts are the class names: for CUB these correspond to bird species, for SUN, the classes describe "scenes" like abbey, seacliff, golf course, and for ImageNet they are the classes like stingray, vulture, water snake, etc. The low-level concepts are: for CUB, class level descriptions of birds such as wing color red, breast color blue, for SUN, descriptions of said scenes such as natural light, swimming and climbing for seacliff, and for ImageNet they are general descriptions of the classes. These sets comprise very diverse top-down textual descriptions allowing for the assessment of the potency of the approach. Can the reviewer elaborate on how this hierarchical construction of descriptions of the classes does not make sense?
>
> > In terms of performance, the proposed CPM underperforms CDM by a considerable margin on both CUB and SUN.
>
> The reviewer makes the argument that CDM outperforms our CPM method by a considerable margin. We apologize if the current layout of Table 1 leads to misinterpretation. Specifically, we do see that the best overall results for CUB and SUN correspond to CDM, but that is using exclusively the high-level concepts (CDM$^H$, that is, the classes themselves) as a concept bottleneck, and it is meant as an ablation for comparison with CPM on the High Level. Within this frame of reference, CDM$^H$ without concept discovery/sparsity exhibits $2.5$% and $2.2$% better accuracy on CUB and SUN respectively. When considering CDM$^H$ with sparsity, this leads to $1.1$% and $0.55$% better accuracy compared to CPM on the High Level. The CDM results that should be considered for the comparison with the Low Level correspond to CDM$^L$, against which CPM (Low Level) provides a very substantial improvement. Finally, it is worth noting that for ImageNet, CPMs exhibit $0.85$% improvement on the high level, $\approx 2$% improvement on the low-level, and $3.2$% when compared to CDM$^H$ that uses the whole concept set, i.e., without discovery. We will clarify this in the camera-ready to prevent this confusion between the different settings.
>
> > Also, it is not fair to claim that label-free CBMs is not capable of concept discovery, since they do automatically discover concepts using CLIP. The only advantage is in terms of sparsity, which can actually be trivially enforced by additional sparsity constraints in label-free CBMs.
>
> When using CLIP, we assess the relationship in terms of cosine similarity of the embeddings of the images and the concepts in the pre-trained CLIP embedding space. We have an implicit measure based on the distance of vectors in that space, that hints about the most similar concepts but in no way explicitly denotes the utility of the concept in the downstream task. At the same time, the Label-Free paper does not follow the same concept-based classification that we follow, but instead uses a pretrained backbone network, followed by a projection layer, aiming to first match the activations of said layer to the concept matrix of CLIP using CLIP-Dissect and then learn a linear classification layer.
> It does indeed consider sparsity, but is not "trivially enforced" as the reviewer would expect. It resorts to a highly non-trivial process, considering a post-hoc elastic-net based sparsity method for the last linear layer, imposed on a class-wise basis, with ad-hoc sparsity-accuracy constraints; at the same time, they do not present the retained number of concepts for each class and do not consider ground truth evaluations. It is not about being "fair" to the method; we never claimed anything or criticized the Label-Free paper, which we found very inspiring for our own work.

---

### Official Review · Reviewer_DfML · 2023-11-01

**Soundness:** 3 good
**Presentation:** 3 good
**Contribution:** 2 fair
**Rating:** 5
**Confidence:** 3

**Summary:**

This paper introduces a hierarchical approach to Concept Bottleneck Models (CBMs) that aims to make decisions comprehensible at multiple levels of granularity. The proposed framework incorporates cutting-edge image-text models with a novel strategy for selecting multi-level concepts, employing Bayesian techniques to induce data sparsity. This method, instead of the conventional whole-image similarity approach, employs a concept hierarchy that targets specific regions of the image for granular concept analysis. The framework's effectiveness is backed by experimental results, showing superiority over existing CBM methods and advancing towards a principled path for interpretable models. The approach emphasizes both low-level and high-level concept discovery, using text and attributes for detailed image patch descriptions and classes for holistic image understanding. A linkage matrix, learned from the data, defines the interplay between these concept levels, resulting in a structured and interpretable model.

**Strengths:**

1-The paper is well-written, offering a clear explanation of the methods used.

2-It provides a well-articulated rationale for the method, with a clear linkage between the objectives and the loss functions employed.

3-The approach demonstrates good performance in identifying and interpreting low-level concepts within images.

**Weaknesses:**

1-The terminology 'hierarchical concept discovery' suggests multiple levels of conceptual granularity, yet the model only delineates high and low-level concepts. To accurately reflect the claimed hierarchy, the model should demonstrate an intermediate conceptual layer, at least within one dataset, to substantiate its hierarchical claims.

2-The distinction between this model and existing transformer-based methods, which also discern image attributes, is ambiguous. Both seem to analyze image patches and assign attributes, making it unclear about the novelty of the proposed approach.

3-The ablative studies provided are not comprehensive, particularly regarding the role and impact of the parameter $\epsilon$. A more detailed examination of this parameter's influence is warranted for a complete understanding of the model's behavior.

**Questions:**

1- Does Figure 1 accurately represent the relationship between high and low-level concepts as described by matrix $B$ in Equation 11 within your experimental results? Or is it to show the overall concept of the method?

2- Regarding Equation 13, could you elaborate on the process for determining the value of the hyperparameter $\epsilon$?

3- Table 1 presents a point of confusion; if CDM and CLIP yield superior classification accuracy, what is the advantage of employing hierarchical concepts as proposed in your methodology?

---

> ### Author Response · Authors · 2023-11-12
>
> We thank the reviewer for highlighting the strengths of our work and for allowing us to further improve on the presentation via the perceptive suggestions/questions.
>
> > The terminology 'hierarchical concept discovery' suggests multiple levels of conceptual granularity, yet the model only delineates high and low-level concepts. To accurately reflect the claimed hierarchy, the model should demonstrate an intermediate conceptual layer, at least within one dataset, to substantiate its hierarchical claims.
>
> As also noted in the response to Reviewer Hcna, at its current formulation, the approach indeed considers only two level of hierarchy. To this end, and taking into consideration the concerns of the reviewer, we will refer to our hierarchical construction as coarse-to-fine, as Rev. Hcna suggested, rather than multi-level. Nevertheless, since we consider more than one level, there exists a hierarchy in the representation, albeit only two layers deep. Expanding to more levels, despite being amenable to an intuitive construction, would require the construction of appropriate concept sets and thorough experimental evaluation and assessment.
>
> > The distinction between this model and existing transformer-based methods, which also discern image attributes, is ambiguous. Both seem to analyze image patches and assign attributes, making it unclear about the novelty of the proposed approach.
>
> We thank the reviewer for raising this point and we'll add a discussion in the camera-ready to explicitly address this distinction. Vision Transformers do indeed split the images into patches and allow the patch representations to interact freely with each other along the model, allowing for rich, context-aware representations, but limiting their interpretability, since it becomes impossible to grasp these interactions. At the same time, it is not straightforward to exploit the low-level hidden spatial information in the context of concept-based interpretability in a principled way.
> In our case, we strongly limit patch interactions by only allowing a single channel of communication that allows the high level concepts to control which low level concepts are detected by providing the appropriate semantic context and aggregate the individual information arising from each patch region.
>
> > The ablative studies provided are not comprehensive, particularly regarding the role and impact of the parameter epsilon. A more detailed examination of this parameter's influence is warranted for a complete understanding of the model's behavior.
>
> We thank the reviewer for pointing out this lack of information about the epsilon parameter. Please find below an ablation study on the effect of the parameter on the final performance of the two levels. We consider the CUB dataset and values of epsilon in [0.0, 0.1, 0.2, 0.5, 0.7, 0.8, 1.0]. When considering ``extreme'' values, i.e., $\epsilon =0$ or $\epsilon=1.0$, the classification performance of the respective levels collapses. This is expected, since there is no classification signal. As the value of $\epsilon$ increases, we observe a gradual improvement in the classification performance of both levels. We posit that the classification signal on the high level allows for the discovery of more discriminant features towards classification while restricting the impact of the sparsity inducing mechanism (to achieve the downstream task). Simultaneously, this provides more flexibility to the low level since more high level concepts are considered, leading to both accuracy improvements and a larger number of low level attributes. This leads to a more expressive flow on information between the two levels, exhibiting the final behavior described in the main text. Finally, we observe that when using values between [0.5, 0.8], the performance is comparable within a margin of error. In this context, equally balancing the classification signal of both levels makes intuitive sense; nevertheless selecting a value in this range to drive the classification potency of the high level provides consistent performance, alleviating the burden of carefully tuning the hyperparameter. We apologise for this omission from the manuscript; we'll include these results and discussion in the camera-ready.
>
> |  $\epsilon$        |         0.0          | 0.1               | 0.2              | 0.5               | 0.7              | 0.8                | 1.0              |
> |:----------:|:----------------:|:-----------------:|:----------------:|:-----------------:|:----------------:|:------------------:|:----------------:|
> | High Level | $0.430 \|\| 10.50$ |  $73.00 \|\| 23.25 $ | $76.20 \|\| 42.50$ | $ 79.20 \|\| 57.00$ | $79.50 \|\| 58.00$ | $79.50 \|\| 57.00 $  | $80.50 \|\| 51.60$ |
> | Low Level  | $62.55 \|\| 21.10$ | $69.50 \|\| 23.50$  | $71.20 \|\| 30.00$ | $72.45 \|\| 31.00$  | $73.00\|\| 33.00$  | $ 72.30 \|\| 31.50 $ | $0.500 \|\| 00.00$ |

---

> ### Author Response · Authors · 2023-11-12
>
> > Does Figure 1 accurately represent the relationship between high and low-level concepts as described by matrix in Equation 11 within your experimental results? Or is it to show the overall concept of the method?
>
> We thank the reviewer for this question, it will help us expand the explanation in the camera ready. Yes, this is an accurate, albeit concise, representation of the relationship between the high and the low-level concepts. For example, seacliff is indeed described by "swimming", "natural light", and "climbing", as is the case with the other connected high level concepts in the Figure. Certainly, there are other attributes describing the high level concepts, but for visualization clarity, we did not include all the attributes.
>
> > Regarding Equation 13, could you elaborate on the process for determining the value of the hyperparameter $\epsilon$?
>
> Please see our previous response.
>
> > Table 1 presents a point of confusion; if CDM and CLIP yield superior classification accuracy, what is the advantage of employing hierarchical concepts as proposed in your methodology?
>
> We agree that the current layout of Table 1 may lead to this confusion, since it includes different settings with varying level of interpretability.
> Classification accuracy is a good metric for assessing the capacity of the network, but not in terms of interpretability. Using only the CLIP embeddings, we do not consider any concepts and subsequently we have no interpretability properties whatsoever.
>
> Our main objective is to introduce a method towards a more principled discovery of attributes describing a given image. Indeed, given that the concept discovery mechanism is active, CDM$^H$ outperform CPM (High-Level) by a small margin in CUB and SUN, while CPM (High and Low) outperform CDM$^H$ on ImageNet in all cases. In our view, there is not a single best method to solve the problem of intepretability. However, in the considered setting, we can see the benefits of the hierachical (coarse-to-fine) construction in the attribute matching case. Compared to CDMs, we observe up to $\approx 60$% relative improvement ($10$% absolute improvement) in attribute matching, validated with ground truth information found in CUB and SUN. We believe that our method provides substantial improvements in this case and opens up the path towards new approaches in the context of concept-based interpretability.
> We will rearrange the camera-ready results in Table 1 to make sure that these different settings are more easily conveyed.

---

> > ### Comment · Reviewer_DfML · 2023-11-22
> >
> > I appreciate the authors' clarifications on the hyperparameter epsilon and the fixed terminology. For the camera-ready version or in future iterations, I recommend presenting the epsilon analysis in a diagrammatic form rather than a table if it provides clearer comprehension. Also in the supplemental, there is a section title separated in text on page 12. Overall, I will maintain my score.

---

> > > ### Author Response · Authors · 2023-11-22
> > >
> > > We thank the reviewer for his response. We have now updated the manuscript to also include a plot of the effect of $\epsilon$ as the reviewer suggested. Please let us know if we could provide any other clarifications or details about our work. If we have already addressed all the concerns of the reviewer raised in your initial review, we would kindly ask if the reviewer could consider adjusting the score taking the rebuttal into account.

---

### Official Review · Reviewer_peBd · 2023-11-06

**Soundness:** 3 good
**Presentation:** 3 good
**Contribution:** 3 good
**Rating:** 6
**Confidence:** 4

**Summary:**

This paper attempts to create interpretable image classifiers with multi-layered, self-discovered concepts. It builds on the work by Panousis et al. (2023), which introduces a way to produce a linear mapping from the space of similarity -- defined as the inner product between the image embedding and the concept embedding -- to the image class. Sparsity is induced with a binary matrix (Z), which also doubles as a representation of concept activation. The work of Panousis et al. (2023) is referred to as the 'high level concept discovery' in this current paper. This paper proposes to split an image into non-overlapping patches and repeat the same method to obtain 'low level concepts' for each patch. A max-pool is applied to predict the image class from the the patch level outputs. The paper then proposes a method to obtain a map between the low level and high level concepts. The joint-model is trained using stochastic gradient variational bayes and the results are compared with other state of the art methods.

**Strengths:**

The paper is well-written and presents an interesting way to discover concepts at multiple levels and link them together. I appreciate that the paper tries to give readers an intuitive overview of the idea presented. Important equations are presented clearly at the right moment to explain the implementation.

**Weaknesses:**

It may be possible to perform out-of-sample evaluation of the model, which the authors did not attempt.

**Questions:**

Suggestions:
1. Please use less italics. There are many words in italics that need not be emphasized and end up being distracting.
2. It may be more standard to use the Hadamard product operator instead of the interpunct to denote element-wise product of matrices.
3. Page 3, Line 1: Shouldn't X_n be a three-dimensional tensor, not 4D? Also, define 'c'. I think it should be the number of channels.
4. Page 3, equation 1: please define 'A'.
5. Page 3, paragraph 7 ("In this work, ...): You overloaded the use of H and L to mean denote High/Low and the cardinality of concept sets. I am just being picky here.
5. Page 3, equation 3: please define A_H
6. Please report the values of the hyperparameters, epsilon and beta, in the main paper. I also spotted that alpha is reported in the appendix but the variable does not appear in the main paper. Does it refer to epsilon?
7. Suppose you have trained on a given set of train data and obtained Z_H, Z_L, W_{Hc} and W_{Lc}. Then, given a set of test data, you can retrain Z_H and Z_L while freezing W_{Hc} and W_{Lc}. By doing so, it may be possible for you to perform train-test evaluation. Can you please affirm or rebut this suggestion? If you think that this is a good idea, can you report the train and test results for your experiments? I would be willing to boost the rating of this paper if this question is addressed.

Questions:
1. How would you decide what are the optimal values of epsilon and/or beta? How would you think about the tradeoff between accuracy and sparsity?
2. In your experiments, you used the classes as the 'high level concepts' and ground truth attributes as the 'low level concepts'. It is conceivable that there is a hierarchy in the ground truth attributes (e.g., leaves, wood --> trees). Is it possible to extend your method to learn this hidden hierarchy?

---

> ### Author Response · Authors · 2023-11-12
>
> We thank the reviewer for his kind comments on the presentation of the approach; we tried to make the work accessible, without overloading the manuscript.
>
> > Suggestions on improvements.
>
> We thank the reviewer for pointing out our errors and suggesting improvement on the presentation. We'll make sure to correct everything in the camera-ready following the reviewer's suggestions/corrections.
>
> > Please report the values of the hyperparameters, epsilon and beta, in the main paper. I also spotted that alpha is reported in the appendix but the variable does not appear in the main paper. Does it refer to epsilon?
>
>  We thank the reviewer for pointing this omission. $\beta$ was set to $1e-4$ in all experiments, while $\alpha$ corresponds to the value of the priors of the latent indicators Z (Eq.(12)). We will clarify this in the revised version. For $\epsilon$, we set the value to $1/2$ in all experiments, but also see the response to Reviewer DfML, where we present an ablation study on the effect of this hyperparameter.
>
> > Suppose you have trained on a given set of train data and obtained $Z_H, Z_L, W_{Hc} and W_{Lc}$. Then, given a set of test data, you can retrain $Z_H$ and $Z_L$ while freezing $W_{Hc}$ and $W_{Lc}$. By doing so, it may be possible for you to perform train-test evaluation. Can you please affirm or rebut this suggestion? If you think that this is a good idea, can you report the train and test results for your experiments? I would be willing to boost the rating of this paper if this question is addressed.
>
> We thank the reviewer for his suggestion and his willingness to improve the score for our work. We were thinking of a similar setup, where we could train the parameters on a single large dataset and combined concept set and test the zero/few-shot capabilities of the model on a different test set. However, the issue that needs to be addressed in both cases is the incompatibility between the concept sets and the number of classes. Another issue that we would be facing with any potential approach that aims to retrain the $Z_H, Z_L$ indicators is that these are driven by two distinct terms: (i) the classification signal, and (ii) the KL divergence. If we remove the classification signal, and indeed we are obligated to do so if we consider a distinct test set, the remaining signal, that is the KL divergence, will push all the indicators towards zero since there is no other term that encourages them to be active. This would warrant a whole new investigation of the method, adjusting for this insufficiency. Nevertheless, the suggestion of the reviewer does provide stimulation and valuable insights on potential avenues that could be explored in order to disentangle this process and make the approach even more generalizable.
>
> > How would you decide what are the optimal values of epsilon and/or beta? How would you think about the tradeoff between accuracy and sparsity?
>
> For the value of $\epsilon$, we just set it to $\epsilon = 1/2$. This decision was motivated by the intuition, that both classification signals from the two levels should equally contribute to the loss (please also see the response to Rev. DfML on the effect on this parameter on the final performance). The $\beta$ parameter is important to avoid the KL term from dominating the task. We tested three values $[1e-2, 1e-3, 1e-4]$ and selected the best performing one (in terms of avoiding oversparsification that leads to accuracy drops) using a dedicated validation set. The trade-off between accuracy and sparsity is an important consideration in sparsity-inducing methods. In our view, a significant advantage of the Variational Bayesian framework is that there is no need to enforce explicit sparsity-accuracy constraints. This leads to a data-driven per example sparsity learned during training instead of a post-hoc sparsification of the weights. At this point, we consider a fixed value, but there are methods that anneal $\beta$ from $0$ to $1$ during training that could potentially remove the burden of selecting the ``best'' value. We aim to explore such methods in the future.

---

> ### Author Response · Authors · 2023-11-12
>
> > In your experiments, you used the classes as the 'high level concepts' and ground truth attributes as the 'low level concepts'. It is conceivable that there is a hierarchy in the ground truth attributes (e.g., leaves, wood --> trees). Is it possible to extend your method to learn this hidden hierarchy?
>
> We thank the reviewer for this interesting question. If we understand correctly, this is closely related to the discussion with Reviewer Hcna and pertains to a bottom-up view of the relation between high and low level concepts. Please also refer to this discussion. One of the motivations of this work was to be able to examine not only the high level concepts but mainly the low level ones. This could potentially allow for drawing conclusions about the high level concept in terms of the uncovered low level attributes. In this context, we can focus on the discovered low-level attributes themselves and reason on the high-level concepts as the reviewer suggests. In our opinion, this is somewhat captured in the proposed framework. Indeed, in the qualitative analyses, we observed that, many times, the discovered low level concepts revealed attributes that are semantically connected to various high level concepts. Moreover, the idea that the reviewer is suggesting could be extremely useful, especially in settings where there is no known hierarchy between the concepts, while in our case, we assume that this relationship is given. This would lead to a very interesting compositional bottom-up approach with no a priori known hierarchies. Within this context, we could potentially devise a method that integrates this bottom-up view, aiming to uncover the hidden hierarchies.

---

### Official Review · Reviewer_Hcna · 2023-11-09

**Soundness:** 3 good
**Presentation:** 3 good
**Contribution:** 2 fair
**Rating:** 5
**Confidence:** 4

**Summary:**

The paper introduces a novel framework for ante-hoc interpretability that leverages image-text models/vision-language models to map unstructured human-understandable concepts to the whole image as well as specific patches of the images. This is in contrast to existing methods which map only the whole image to the text concept through similarity metrics. Further, the paper also unfolds a paradigm in concept discovery for enabling multi-level concept selection. The idea of the work stems from the notion that current concept activations in CBM methods tend to activate generic low-level concepts (class attributes: whiskers, eyes, etc.) that represent a high-level concept (the class present in an image: elephant, cat, etc.) , even when some low-level concepts (eg: beak of birds, tusks in elephants, etc.) may not be present/visible in that particular image. This may lead to significant concept omission, thus motivating the hierarchical multi-level concept discovery approach proposed herein.

**Strengths:**

+ The paper clearly identifies the existing problems and drawbacks of CBM models as stated through points (i) to (iv) in section 1.
+ A potential problem with current CBM models and inability to consider a flexible concept bottle-neck layer that is inherently hierarchical, is interesting and does open a new problem in concept-based interpretable methods.
+ The proposed metric for evaluating interpretation capacity is also to be encouraged for future works in this direction: it helps to address the importance and quality of concepts in the final prediction.
+ The work effectively utilizes a variational inference approach to estimate the presence of high-level and low-level concepts, the method seems convincing.
+ The baselines considered for the experiments are well-thought through.
+ The paper is generally easy to follow and well-written.

**Weaknesses:**

* At a high level, I am not completely convinced by the claim of "hierarchical multi-level concept discovery", since the methodology by design is intended only for two levels of concepts, not any deeper. It'd have been better to say "two-level" or "coarse-to-fine concepts", instead of "hierarchical multi-level".
* Sec 3.3 talks about how the high-level and low-level concepts are connected. It states that the presence of a high-level concept provides information on which low-level concepts may or may not exist. I would have expected to see the concepts organized the other way – the presence of a set of low-level concepts triggers a high-level concept, the way it would happen in a visual processing pathway. Why is this flipped? What are the implications of this flip?
* Continuing with the above point, if the high-level concept is first inferred, can’t the low-level concepts be inferred in an independent manner without the network itself? Acc to Fig 1, there is an independent classification layer for high-level and low-level concepts anyway, and one could use just the high-level module to classify.
* The need for hierarchical concept selection could be strengthened by designing experiments in CPM setting with A_Hactivations and without (replace it with whole set) to see the performance of A_L since the proposed model relies on both activations. Such ablation studies and detailed analysis seems lacking. For another example, experiments on different size of concept-pool would also help in evaluating the overall performance of the method across a varied condition resulting in validating the findings better. As stated earlier, including more levels of concept would be needed to strengthen the idea presented as a “multi-level concept discovery” work.
* Technically speaking, the work seems to be a derivative of the cited paper “Sparse Linear Concept Discovery Models”; the architectural differences lie in the use of two-level of concepts: high and low, rather than a single set of concepts. Furthermore, important ideas have been drawn from the paper such as sparsity inducing Bayesian approach.
* The idea of multi-level concepts can be introduced directly in a standard CBM too, such that high-level concept activations in the first would trigger the low-level concept activations (with end-to-end). Why will this not address the stated purpose?
* It is not clear why Region-CLIP or other CLIP variants that deal with images at the level of patches/superpixels/regions were not considered for the second level of concepts.
* The literature survey does not seem complete; there are papers like “Probabilistic CBMs, ICML 2023” that have been missed.


*Minor suggestions:*
* I would recommend using \citep for references used as part of a sentence vs \cite in another places. This helps the readability of the paper.
* In Eqns 1 and 3, A seems undefined. It appears that it should be \mathbb{A}. Please correct this.

**Questions:**

1. The paper, in its introduction, lists limitations of CBM models. It is not clear why "(iii) their interpretability is substantially impaired due to the sheer amount of considered concepts, and (iv) they are not suited for tasks that require greater granularity." are a problem for CBMs. How does having a large number of concepts actually affect CBMs? I found this argument to be rather weak.
2. A similar statement is also made in Sec 2: "With the number of concepts ranging from the 100s to the 1000s, this can severely undermine the sought-after interpretability." Why do a large number of concepts affect interpretability?

---

> ### Author Response · Authors · 2023-11-12
>
> We thank the reviewer for his insightful comments and questions regarding our approach. At the same time, we are very delighted with the recognition of the strengths of our work from the reviewer, especially concerning the introduction of a new problem in concept-based interpetability and the use of the Jaccard metric for assessing the quality of the obtained relations.
>
> > At a high level, I am not completely convinced by the claim of "hierarchical multi-level concept discovery", since the methodology by design is intended only for two levels of concepts, not any deeper. It'd have been better to say "two-level" or "coarse-to-fine concepts", instead of "hierarchical multi-level".
>
>
> Indeed, the proposed methodology at this stage is formulated for and only considers two levels of hierarchy. However, we posit that it is possible to construct deeper concept hierarchies and connect them via the corresponding binary indicators. This could be achieved in various ways using different views for the different levels, e.g. satellite images, ground-level images and segmented views, and hierarchies of concepts, e.g., forests, trees, tree attributes. We agree nevertheless that at this point, the method does not address this setting; we'll use the coarse-to-fine wording in the camera-ready per the reviewer's suggestion.
>
> > Sec 3.3 talks about how the high-level and low-level concepts are connected. It states that the presence of a high-level concept provides information on which low-level concepts may or may not exist. I would have expected to see the concepts organized the other way – the presence of a set of low-level concepts triggers a high-level concept, the way it would happen in a visual processing pathway. Why is this flipped? What are the implications of this flip?
>
> We thank the reviewer for raising this interesting aspect. Indeed there are two ways to approach this setting: (i) top-down (which we consider), and (ii) bottom-up (which the reviewer suggests). In our view, a combined bottom-up-then-top-down approach is what would most closely follow a human-like behavior when analysing an object. However, it is the second step that is more of a conscious process: we first become aware of the whole object, e.g., a bird or a dog, even if we have subconsciously perceived a lot of low-level cues to reach that conclusion, and then, based on this high-level knowledge, we can draw further conclusions about the nature of the lower-level image characteristics, e.g. interpreting a furry texture as either feathers or fur. In a purely bottom-up approach, we would first analyse the low-level characteristics, such as shapes and textures, and we would then try to reason about the whole context in order to assign them semantics, e.g. ears, tail, fur. In our opinion, there isn't a single right approach for solving this problem in the context of interpretability.
>
> We posit however, that the information exchange that takes places between the high and the low levels via the learning process of the binary indicators does indeed allows for context information sharing between both levels (in the forward pass only from high to low, but also the inverse during training).
>
> This behavior is also suggested by the experimental evaluation, where we activate features from different high-level concepts, significantly altering the patterns of concept activations of the two levels compared to when individually training the levels.
> We will stress these concepts in the camera ready version.

---

> ### Author Response · Authors · 2023-11-12
>
> > Continuing with the above point, if the high-level concept is first inferred, can’t the low-level concepts be inferred in an independent manner without the network itself? Acc to Fig 1, there is an independent classification layer for high-level and low-level concepts anyway, and one could use just the high-level module to classify.
>
> We preemptively apologise in case we missed the point that the reviewer makes, since in our view, we believe that there are two interpretations. The first interpretation is to first independently infer the high-level concepts and then use these learned concept activations and learn the low-level attributes in an independent way. We posit that this defeats the purpose of end-to-end training and specifically the information exchange between the two levels. As we have shown, linking the two levels has a significant impact on the behavior of both levels, both classification wise but also for the concept retention rates. At the same time, this will greatly constrict what the low-level can learn, since it is now forced to use the inferred high-level concepts in a ``deterministic'' way, not allowing the discovery of potentially contributing attributes from other high-level concepts (as was the case with the Sussex Spaniel analysis), thus also not allowing the alteration of the originally inferred high level concepts. The second interpretation of this point is to use a single classification loss only from the high level of the framework. We posit that this will obstruct the low level from finding discriminative low level attributes, hindering the performance of the low-level module and diminish the feedback signal between the two levels. Please also see the response to Rev. DfML, where we set $\epsilon=1.0$ and observe this exact behavior. The intuition behind our construction was to allow for concept analysis and strong performance on both levels so that we could investigate the decision making process of the network both individually and synergistically. This also ties to the response to the next question of the reviewer.
>
> > The need for hierarchical concept selection could be strengthened by designing experiments in CPM setting with A\_H activations and without (replace it with whole set) to see the performance of A\_L since the proposed model relies on both activations. Such ablation studies and detailed analysis seems lacking. For another example, experiments on different size of concept-pool would also help in evaluating the overall performance of the method across a varied condition resulting in validating the findings better. As stated earlier, including more levels of concept would be needed to strengthen the idea presented as a “multi-level concept discovery” work.
>
> We thank the reviewer for this comment, since it allows us to clarify the settings. The reviewer suggests that we should design experiments with and without $A_H$ to assess performance. We would like to note that these experiments are already included. Specifically, the full CPM setting considers the $A_H$ activations as described in the main text. At the same time, the CDM$^L$ setting is exactly equivalent to not using the $A_H$ activation mechanism, i.e., uses the whole pool of low-level attributes $A_L$ since there is no information about the high level concepts; this directly translates to the setting where all high-level concepts are active. Since we do not consider the data-driven binary indicators for the high level (thus using the whole concept set), there is no flow of information between the high and the low-level and the levels are trained independently. This leads to very poor performance in the low-level. Only by activating $Z^H$ it is possible to consider information exchange between the levels. We apologize for the lack of clarity on this and will make sure to expand on this and the previous point of the reviewer in the camera-ready.
>
> Constructing an appropriate concept pool is a challenging task in the context of all concept-based methods. We believe that it's not solely about the quantity/size of the concept pool, but mainly its quality. We aim to explore how we can effectively create consise hierarchical concepts, but this does not undermine the efficacy of the approach. Indeed, we have seen in the qualitative analysis that the concept sets describing each class of ImageNet (taken from 20 random descriptions per class from Yang et. al, (2023)) can be of varying quality. Nevertheless, our framework manages to exploit information from different classes to match the presented input. At the same time, for the SUN and CUB datasets, we chose to use the already existing low-level attributes; this allows for exploiting the ground truth information during the evaluation and measure the attribute prediction potency of the proposed framework.

---

> ### Author Response · Authors · 2023-11-12
>
> > Technically speaking, the work seems to be a derivative of the cited paper “Sparse Linear Concept Discovery Models”; the architectural differences lie in the use of two-level of concepts: high and low, rather than a single set of concepts. Furthermore, important ideas have been drawn from the paper such as sparsity inducing Bayesian approach.
>
>  Indeed, this work draws inspiration from the mentioned paper. We state this in the introduction and in Section 3.1. The novelty of the method lies on the introduction and assessment of method in the patch input scenario (along with the aggregation of the information), the introduction of the concept hierarchy, the linkage between the high and the low level views of the image, and the introduction of the Jaccard measure as a measure of efficacy in terms of interpretability.
>
> >The idea of multi-level concepts can be introduced directly in a standard CBM too, such that high-level concept activations in the first would trigger the low-level concept activations (with end-to-end). Why will this not address the stated purpose?
>
> In our view, it is not straightforward to see how the hierarchical process can be incorporated in conventional CBMs is a principled way. Indeed, CBMs commonly rely on the (cosine) similarity between concepts and images to assess the relationship between the two, without *explicitly* denoting which concepts are triggered/activated or not. So it is not clear, which high level concepts are activated in the context of the downstream task and which concepts in the low level should be considered. One way to approach this could potentially by taking the top-k most similar concepts or devise a sampling procedure based on the similarity values. This leads to a totally different approach whose potency and validity need to be explored. We posit however, that our approach, based on binary indicators modeled via solid Variational Bayesian arguments, yields a more principled construction.
>
> > It is not clear why Region-CLIP or other CLIP variants that deal with images at the level of patches/superpixels/regions were not considered for the second level of concepts.
>
> There exists a variety of ways to approach the second level of the hierarchy. Indeed, as the reviewer suggests, clip variants such as Region CLIP could be highly appropriate to assess and even improve performance. In our setting, we explored the potency of the hierarchical construction based on the conventional CLIP encoder, and if low level information could be captured with a simple intuitive process in a principled way. We believe that our work can pave the way for future explorations and ``open a new problem in concept-based interpretable methods'' as the reviewer aptly notes.
>
> > The literature survey does not seem complete; there are papers like “Probabilistic CBMs, ICML 2023” that have been missed.
>
> We thank the reviewer for the related literature. Indeed, we missed this work and we'll make sure to include a discussion in the camera-ready. This method however, is mainly focused on uncertainty estimation via probabilistic embeddings, exhibiting substantial classification performance gap compared to our approach.
>
> > The paper, in its introduction, lists limitations of CBM models. It is not clear why "(iii) their interpretability is substantially impaired due to the sheer amount of considered concepts, and (iv) they are not suited for tasks that require greater granularity." are a problem for CBMs. How does having a large number of concepts actually affect CBMs? I found this argument to be rather weak.
>
> We thank the reviewer for pointing out this insufficient phrasing. We posit that *analysing* (and not considering) thousands of concepts in the context of concept-based interpretability is in our view counter-intuitive. Indeed, in conventional non-sparse CBM approaches, when we want to assess the relationship between an image and a set of concepts, we need to examine the image's similarity to the *whole set* of concepts. This constitutes an arduous and time-consuming process, while at the same time relying only on the cosine similarity between the modalities thus, not allowing for an explicit reasoning of which concepts were used for the downstream task. It is not a matter of how many concepts does one consider for the CBM; it is how you select a meaningful and human interpretable subset of these potentially thousands concepts in a principled manner. We'll rephrase this argument to match the aforementioned view.
>
> > A similar statement is also made in Sec 2: "With the number of concepts ranging from the 100s to the 1000s, this can severely undermine the sought-after interpretability." Why do a large number of concepts affect interpretability?
>
> Please see the response to the previous question.

---

### Author Response · Authors · 2023-11-20
**Updated Manuscript**

Dear AC and Reviewers,

thank you again for all your comments and important feedback.
We have now updated our manuscript according to the reviewers' suggestions.

Any further comments are gladly welcomed.